# Enhanced Targeted Delivery of Minocycline via Transferrin Conjugated Albumin Nanoparticle Improves Neuroprotection in a Blast Traumatic Brain Injury Model

**DOI:** 10.3390/brainsci13030402

**Published:** 2023-02-25

**Authors:** Venkatesan Perumal, Arun Reddy Ravula, Agnieszka Agas, Aakaash Gosain, Aswati Aravind, Ponnurengam Malliappan Sivakumar, Shanmuga Sundari I, Karthik Sambath, Sivakumar Vijayaraghavalu, Namas Chandra

**Affiliations:** 1Center for Injury Biomechanics, Materials and Medicine, Department of Biomedical Engineering, New Jersey Institute of Technology, Newark, NJ 07102, USA; 2Institute of Research and Development, Duy Tan University, Da Nang 550000, Vietnam; 3School of Medicine and Pharmacy, Duy Tan University, Da Nang 550000, Vietnam; 4Computational Biology Special Lab, Department of Biotechnology, Bannari Amman Institute of Technology, Sathyamangalam 638401, India; 5Department of Chemistry and Environmental Science, College of Science and Liberal Arts, New Jersey Institute of Technology, 323 Martin Luther King Jr. Blvd., Newark, NJ 07102, USA; 6Department of Life Sciences (Zoology), Manipur University, Imphal 795003, India

**Keywords:** minocycline, nanoparticle, albumin, HPLC, desolvation method, targeted delivery, biodistribution

## Abstract

Traumatic brain injury (TBI) is a major source of death and disability worldwide as a result of motor vehicle accidents, falls, attacks and bomb explosions. Currently, there are no FDA-approved drugs to treat TBI patients predominantly because of a lack of appropriate methods to deliver drugs to the brain for therapeutic effect. Existing clinical and pre-clinical studies have shown that minocycline’s neuroprotective effects either through high plasma protein binding or an increased dosage requirement have resulted in neurotoxicity. In this study, we focus on the formulation, characterization, in vivo biodistribution, behavioral improvements, neuroprotective effect and toxicity of transferrin receptor-targeted (tf) conjugated minocycline loaded albumin nanoparticles in a blast-induced TBI model. A novel tf conjugated minocycline encapsulated albumin nanoparticle was developed, characterized and quantified using a validated HPLC method as well as other various analytical methods. The results of the nanoformulation showed small, narrow hydrodynamic size distributions, with high entrapment, loading efficiencies and sustained release profiles. Furthermore, the nanoparticle administered at minimal doses in a rat model of blast TBI was able to cross the blood–brain barrier, enhanced nanoparticle accumulation in the brain, improved behavioral outcomes, neuroprotection, and reduced toxicity compared to free minocycline. Hence, tf conjugated minocycline loaded nanoparticle elicits a neuroprotective effect and can thus offer a potential therapeutic effect.

## 1. Introduction

Traumatic brain injury (TBI) is an acquired injury from an exterior force that is associated with long-term cognitive deficits related to trauma-induced neurodegeneration [1,2]. Approximately 1.7 million head injuries occur annually in the United States, resulting in 282,000 hospitalizations and over 50,000 fatalities. In soldiers, blast explosions are the primary cause of TBI, accounting for 50 percent of all injuries [3], and the number of blast TBI (bTBI) cases has escalated in the past 20 years [4,5,6]. According to previous literature, bTBI can cause secondary injury mechanisms such as oxidative stress, neuroinflammation, and neurodegeneration and severely affect the quality of life of military personnel returning from combat zones [3]. Neuroinflammation is a prominent feature associated with various forms of TBI including bTBI [7]. Thus far, more than 30 phase III clinical trials to treat TBI have failed [8,9,10], including the use of broad-spectrum antibiotics such as minocycline. One of the major reasons contributing to this is most small molecular weight drugs possess high protein binding and consequently face rapid elimination by hepatic metabolism and renal excretion, resulting in a short half-life (small molecular drugs) in circulation and a limited percentage of the drug being able to reach the brain [11,12]. Moreover, the high dose level required for neurological treatment causes harmful side-effects and amplified neurotoxicity, vestibular dysfunction and other chronic neural damage [13,14,15].

Minocycline is a semi-synthetic, broad-spectrum tetracycline antibiotic that has been available for clinical use since 1971 [16]. Due to its anti-inflammatory and neuroprotective properties [17,18]. Minocycline’s neuroprotective role has been studied in various experimental disease models, such as Alzheimer’s, Parkinson’s and Huntington’s diseases, as well as focal cerebral ischemia, multiple sclerosis, traumatic brain injury and spinal cord injury [18,19,20,21]. This antibiotic has also been reported to ease neuro-inflammation and guard the blood–brain barrier (BBB) in ischemic stroke [22,23,24]. While a standard dose of 3 mg/kg is sufficient to treat infectious and inflammatory diseases, a repeated high dose of approximately 20–100 mg/kg is required to produce a neuroprotective effect; which leads to harmful complications and aggravates neurotoxicity [25]. Additionally, minocycline possesses limited aqueous solubility (30–100 mg/mL in water), high plasma protein binding (up to 76%), a short half-life, and low BBB permeability [26], making the delivery and bioavailability of therapeutic concentrations, particularly at the desired site of action, a challenge [27]. Given the toxicity and minimal half-life of minocycline, nanotechnology-based drug delivery is the preferred treatment method, as it can evade many complications that arise from direct drug administration [28,29,30,31,32]. In particular, the BBB is transiently compromised in blast-induced TBI; hence, a non-targeted nanoparticle can be administered before the BBB reestablishes its integrity [33,34]. However, after the repair of the BBB, this becomes no longer possible and a targeted approach is required. Binding the nanoparticle with ligands would enable it to engage in receptor-mediated transcytosis and traverse the BBB, resulting in the accumulation of the nanoparticle in the brain parenchyma. Therefore, it is critical to overcome the side effects of minocycline at high doses and use targeted delivery to improve its biodistribution and to enhance the bioavailability and therapeutic efficacy of minocycline. 

Transferrin receptors have reported high expression levels exclusively in the brain endothelium as compared to the peripheral endothelium [35,36]. Prior studies have shown apolipoprotein E conjugated nanoparticles facilitating drug transport across the blood–brain barrier via surface lipoprotein receptors on the brain capillary endothelial cell membranes [37]. Anja Zensi et al. have corroborated these findings by uncovering that the cerebral endothelium uptakes nanoparticles covalently bonded to apolipoprotein E via endocytosis followed by transcytosis into the brain parenchyma [38]. Another study found similar results with the covalent binding of transferrin (tf) or OX26/R17217 antibodies to loperamide-loaded albumin nanoparticles as they were able to deliver the drug to the BBB and induced significant anti-nociceptive effects in a tail-flick test in ICR (CD-1) mice following intravenous injection [39,40]. A number of targeted delivery approaches have been explored for the treatment of various brain disorders [41]; therefore, nanoparticles covalently coupled with ligands (e.g., Transferrin) offer the advantage of delivering drug molecules across the BBB through receptor-mediated uptake by brain capillary endothelial cells and localization at brain parenchymal injury sites (as shown in Figure 1). 

The major issues that plague drug delivery methodologies to treat CNS diseases can be attributed to a failure to achieve sufficient drug concentrations at the target site and release of the drug in a continuous manner for extended periods of time [42]. An ideal nanocarrier for drug delivery across the BBB should possess a size of <200 nm, have reduced toxicity, an extended circulation time, a controlled release profile, be biodegradable and have biocompatible characteristics [43,44]. In general, nanoparticle formulations range from using natural or synthetic polymers to using lipids and metallic materials [45,46,47]. Polymers have been extensively applied preclinically and at the clinical level as a drug vehicle in various disease models and human disease conditions [48]. Natural polymers originate from proteins such as albumin, collagen and gelatin or from polysaccharides such as alginate, chitosan, and dextran [47,49,50]. Nanoparticles formulated with albumin polymers offer the benefits of ease of chemical modification, high drug loading, entrapment capacity, non-immunogenicity, biocompatibility, and cryoprotectant capability as a shelf-stable nanomedicine [44,51]. Because of albumin’s long half-life (19 days), an albumin-based nanocarrier can also increase the plasma area under the curve (AUC) value and sustain systemic concentrations for a relatively extended time [52]. Furthermore, albumin can bind to a wide variety of therapeutic molecules. Various surface modifications can also be performed to enhance drug–protein binding and promote drug-targeting [53]. The first FDA-approved nanotechnology-based albumin nanocarrier (Nab-paclitaxel, i.e., Abraxane) was formulated and marketed for human use in the treatment of metastatic breast cancer, pancreatic cancer and non-small cell lung cancer [54,55].

The protein NPs were obtained by the well-established desolvation procedure [56] which involves the exact addition of a desolvating agent (ethanol) to an albumin solution at an ideal pH (optimal size and encapsulation) with continuous mixing up until turbidity is accomplished (As shown in Appendix A). A decline in the solubility of albumin resulted in phase separation in water during the desolvation process advancing to nanoparticle growth. Additionally, the nanoparticles can become stable by crosslinking the lysine and guanidino side chains of albumin with the crosslinker glutaraldehyde [57]. Once crosslinking is increased, the stiffness of the nanoparticle results in decreases in particle size due to the formation of more dense particles. In our study, the hydrophobic minocycline aggregates in an aqueous solution and interacts with the hydrophobic regions of BSA to form the nanoparticle, which is then stabilized with glutaraldehyde by cross-linking the surface amino groups of BSA [58]. When the NH_2_ from the BSA molecule comes in contact with the carbonyl group of glutaraldehyde, it reacts to form an -N=C- bond by losing a molecule of water. This results in the formation of nanoparticles of albumin crosslinked with glutaraldehyde [59]. Even though glutaraldehyde crosslinked with albumin has no toxicity and the obtained NPs are non-toxic [60], free, unbound glutaraldehyde may display toxicity and it is for these reasons that the unreacted glutaraldehyde was removed by centrifugation and purification. The objective of this study is to offer knowledge on the bioavailability and neuroprotection needed following brain injury with a significantly reduced drug concentration while achieving maximum bioavailability of minocycline through a novel minocycline-loaded nanoparticle (MANP). Our proposed research of tfr (transferrin receptor)-targeted MANP nanoparticles will exploit both receptor-mediated endocytosis via the vasculature and a transiently leaky BBB following blast TBI for enhanced bioavailability at the injury site and reduced toxicity of the drug [33,61]. 

The objective of the present study is to formulate and characterize tf-conjugated minocycline-loaded bovine serum albumin (BSA) based nanoparticles on a rat TBI model by studying in vivo biodistribution, behavioral changes, and other neuroprotective effects. In our study, BSA was used as the nanocarrier as it possesses high stability with low toxicity and immunogenicity [60,62]. To the best of our knowledge, this is the first study to report a novel tf (transferrin)-conjugated albumin-based nanoparticles formulation to achieve enhanced brain delivery of minocycline at a reduced dose in a brain injury model. Further in vivo biodistribution studies performed in a rat TBI model were also studied to validate the enhanced bioavailability of minocycline in the brain (as shown in Figure 1). 

## 2. Materials and Methods 

### 2.1. Chemicals, Reagents and Instruments

Minocycline, Bovine Serum Albumin (lyophilized powder, ≥96% agarose gel electrophoresis), anhydrous ethanol (200 proof, anhydrous, ≥99.5%) crosslinker NHS-PEG-MAL-5000 and transferrin (human: minimum 98%) were purchased from Sigma Aldrich Saint Louis, MO, USA. HPLC Grade Ammonium acetate was purchased from VWR. HPLC-grade water, Methanol and Acetonitrile were purchased from Fisher Scientific. HPLC-grade glutaraldehyde (100%) was obtained from Alfa Aesar (Ward Hill, MA, USA). The Millipore Milli-Q Plus apparatus was used to procure ultrapure water. All other chemicals used in this study were of analytical grade. Traut’s reagent (2-Iminothiolane) and Ellman’s reagent were obtained from Pierce (Rockford, IL, USA). The PD-10 Columns SephadexTM G-25 M were from GE Healthcare. Lyophilizer, Centrifuge (Effendorf centrifuge 5810R, Thermo scientific sorvall RC 6+, Waltham, MA, USA), Zeta sizer (malvern), HPLC (Thermo Fischer), Sonicator and Rota evaporator (Rotaevapor Buchi, R-210) instruments were used in the study.

### 2.2. Formulation of tf Conjugated Minocycline Loaded Albumin Nanoparticle (tf-MANP)

The tf-conjugated BSA nanoparticle was formulated in three steps [37,39]. In the first step, minocycline-loaded albumin nanoparticle (MANP) was prepared using a modified desolvation method [57,63]. Briefly, 10% BSA (*w*/*v*) in HPLC-grade water was stirred (at 600 rpm) with 7.5% of minocycline (*w*/*v*) at room temperature for drug absorption onto albumin. After 1 h of continuous stirring, the pH value was adjusted from 7.5 to 8.5 using 0.1 M NaOH. The mixture was then desolvated through the addition of a suitable amount of ethanol, using a peristaltic pump at a rate of 1 mL/min while stirring (at 600 rpm). Ethanol addition was sustained till turbidity point and residual ethanol was removed by a rotary evaporator at 4 °C. Then, the formulated minocycline-loaded nanoparticle was stabilized by crosslinking with 8% glutaraldehyde solution for 24 h. The nanoparticle in solution was ultra-centrifuged (Sorvall LYNX 6000, Superspeed Centrifuges) at 36,288× *g* force for 40 min. 

2-iminothiolane solution was then added to bind a sulfhydryl group to the transferrin and was quantified through the use of Ellman’s reagent. Briefly, transferrin was dissolved in phosphate buffer (1 mg/mL at pH 8.0) and incubated with 12.8 µL (50.85-fold molar excess) of 2-iminothiolane solution (6.9 mg in 1.0 mL phosphate buffer, pH 8.0) in the dark for 2 h at 20 °C under constant shaking (500 rpm). Thereafter, the thiolated transferrin was purified by PD-10 Columns SephadexTM G-25 M, using phosphate buffer (pH 8.0) as an eluent.

Subsequently, NHS-PEG-MAL-5000 solution in 10-fold molar excess was introduced to the nanoparticles to cross-link activate them. To conjugate the NPs, 500 µL of thiolated and purified transferrin solution was added to 500 µL of reactive BSA NPs. The mixture was incubated while shaking for 24 h at room temperature. Thiolated transferrin excess was removed by 2-fold nanoparticle centrifugation, redispersed in water and lyophilized. Precipitate obtained from the centrifugation was washed with pure water three times and then freeze-dried with the addition of 50 mg mannitol to obtain a brownish fine powder of tf conjugated MANP. For further characterization, a stock suspension of NP was used. Similarly, a drug-free ligand conjugated empty albumin nanoparticle (tfEANP) was also prepared.

Two separate approaches were utilized to redisperse the lyophilized MANP, physical vortexing and sonication [64]. Manual shaking was applied using weighed quantities of lyophilized NP with phosphate buffer saline pH 7.4. The nanosuspension was subject to gentle shaking for 2 min to redisperse the solution and then immediately measured for particle size using a Malvern zetasizer. Micrometer-sized particles were considered non-dispersible. A Sonication method was applied with lyophilized NP in phosphate buffer saline pH 7.4 for 2 min using a bath sonicator for redispersibility.

The conjugation of Cy5.5 to BSA nanoparticles: The near-infrared (NIR) fluorophore Cy5 was labeled on the albumin nanoparticle to investigate the in vivo distribution of NPs in brain injury [65,66,67]. BSA nanoparticles were conjugated to Cy5.5 according to the manufacturer’s recommended protocol [65,66]. Briefly, a reaction between BSA and the Cy5.5-NHS ester (Invitrogen, Carlsbad, CA, USA) at a molar ratio of 1:2 was performed in the dark at room temperature for 1 h. Unconjugated dye was removed by dialysis against phosphate-buffered saline (PBS) using a Slide-A-Lyzer membrane cassette (3500 MWCO) for 24 h at room temperature. Dialyzed samples were filtered through a 0.2-μm syringe filter to ensure quality before use.

### 2.3. Characterization of the Nanoparticles

Mean particle size, particle-size distribution and zeta potential of the nanoparticles were measured using Zeta sizer (malvern). Particle size and zeta potential measurements were conducted on freshly prepared dispersions of nanoparticle stock suspension (1 mg/mL) in PBS at pH 7.4. The results were reported as average values from triplicate runs of three independent experiments for each sample. The structural morphology of the nanoparticle was determined by field emission SEM (LEO 1530VP). For SEM imaging, nanoparticles in ethanol (20 μL) were placed and converted into powder on the surface of carbon substrates [60,68]. Prior to analysis, samples were placed onto metal stubs using double-sided adhesive tape, and sputter coated with gold/palladium to make them electrically conductive and suitable for SEM imaging (EM JSM-7900F, JEOL). For TEM analysis, 10–20 μL of the NP solutions in ethanol were placed on a 200-mesh copper grid coated with carbon. The copper grid was allowed to dry for 2 h at room temperature before imaging (JEM-F200/F2). The FT-IR transmittance spectra of the nanoparticles were obtained using Spectrum 100 FTIR spectrometer (PerkinElmer, Waltham, MA, USA) [60,68,69]. Data collected in the wavenumber range of 400–4000 cm^−1^ at a resolution of 1/cm was measured. 1H NMR spectra of the BSA, tfMANP and cy5.5 tagged MANP were recorded on a Bruker Avance II 400 MHz spectrometer in deuterated chloroform (CDCl3) at room temperature.

To determine the characteristic preservation of the alpha-helical secondary structure, circular dichroism (CD) spectra of BSA, tfMANP and cy5.5 tagged MANP were recorded at a concentration of 500 µg/mL in PBS buffer using a Dichroism Spectropolarimeter (J-810, Jasco International Co., Ltd., Easton, MD, USA) [70,71]. The far UV region was scanned between 190 nm and 240 nm. All the data were buffer corrected and converted to molar residual ellipticity (MRE). MRE was calculated by [*θ*] =$\;\frac{\theta xm}{cxnrxl}$ where *θ* is MRE in millidegrees, m is molecular weight in g/mol, c is concentration in mg/mL, l is the pathlength of the cuvette in cm, and nr is the number of amino acids in the protein.

### 2.4. Entrapment Efficiency and Loading of Minocycline in tf Conjugated MANP 

The amount of minocycline entrapped in the albumin nanoparticle was quantified using a validated HPLC method. The quantity of the drug encapsulated was measured through indirect means, by quantifying the amount of minocycline in supernatant obtained after ultra-centrifugation. Further, the supernatant was dialyzed (20 kDa) to remove the traces of proteins. Briefly, 5 mL of supernatant was transferred into a dialysis cassette (10 mL, cutoff 20 KDa) and then the cassette was placed in 200 mL of beaker with HPLC grade water. Before putting the sample into a dialysis cassette, the membrane of the cassette was hydrated with water. The media was stirred at 250 rpm overnight and 1 mL of dialysis media was withdrawn for the HPLC analysis vial [63,72].
$$\%\;Encapsulation\;efficiency=\frac{\left(total\;minocycline–minocycline\;in\;the\;supernatant\right))}{total\;minocycline}\times 100$$

For minocycline loading efficiency, known quantitative albumin nanoparticle was dispersed in purified water and added as needed. The nanoparticle suspension was sonicated for 30 min and filtered through a 0.25-μm membrane (Whatman^®^ membrane filters nylon pore size 0.2 μm, diam. 25 mm) and quantified by HPLC method [72].
$$Drug\;Loading\;R=\frac{\left(Weight\;of\;minocycline\;in\;nanoparticles\right)}{Total\;weight\;of\;nanoparticle}\times 100$$

### 2.5. In Vitro Minocycline Release from Nanoparticle

The in vitro release profile of minocycline from the tf-conjugated MANP was investigated by a dialysis method using phosphate-buffered saline (PBS; 0.01 M, pH 7.4) as the release medium. Briefly, 1 mL of minocycline solution or tf-conjugated MANP suspension (1 mg/mL) in PBS was added in a dialysis bag (MWCO 80,000 Da) and incubated in 200 mL of release medium at 37 °C at the shaking speed of 100 rpm. For certain time points, a 0.5 mL aliquot was withdrawn (0 to 72 h) and replaced with an equal volume of fresh-release medium. The samples were then subject to HPLC analysis, as described above, and shielded from light exposure during the process. These samples were analyzed in triplicate. 

### 2.6. Stability of tf Conjugated MANP 

A stability study was performed using lyophilized tf-conjugated MANP stored at 4 °C and 25 °C at predefined timepoints (0, 1, 3, 5, 7, 14, 21, 28, 60 and 90 days) [73,74]. Lyophilized minocycline-loaded albumin nanoparticles were suspended with 1 mL of PBS, and all the samples were analyzed for changes in particle size, polydispersity index (PDI) and surface charge. 

### 2.7. Neuroprotection, In Vivo Biodistribution and Toxicity of Targeted tf-MANP in Blast TBI Rat Model

*Animals:* Ten-week-old male Sprague-Dawley (Charles River Laboratories, Wilmington, MA, USA) rats at 350 ± 50 g were used in accordance with protocols approved by Rutgers University Institutional Animal Care and Use Committee (IACUC approval: PROTO201900142). The animals were housed with free access to food and water in a 12-h dark-light cycle at 22 °C. Rats were divided into five groups (*n* = 6); sham controls, blast and three treatment groups of animals exposed to a moderate blast of 180 kPa) [18,75].

*Blast injury:* Rats were subjected to a single blast wave as described previously in (https://doi.org/10.3390/micro3010008, accessed on 19 February 2023) [33,61,76,77]. Rats were then strapped securely to an aluminum plate using a cotton cloth wrapped around the body and placed horizontally inside a 6 m long and 9 in. cross-section shock tube, located 2.8 m away from where the shockwave was generated (as shown in the Appendix A). The cloth provided no protection against the shockwave but prevented excessive motion of the head. The pressure waveform was recorded using PCB Piezotronics sensors model 134A24 (Depew, NY, USA) at 1.0 MHz sampling frequency for a duration of 5 ms. Rats were exposed to a single moderate shock wave at 180 kPa. Sham control rats received anesthesia and noise exposure without being subjected to the blast, i.e., anesthetized animals were placed next to the shock tube when the blast wave was generated. Following blast injury, modified neurological severity score (NSS) was evaluated.

*Treatment regime study plan for biodistribution:* Targeted nanoparticles and free minocycline were intravenously injected into the rats at a dose of 3 mg/kg, 4 h single moderate blast (180 kPa) exposure. Samples were collected after 3 h and 24 h for minocycline and nanoparticle-treated groups, respectively. The time points for the euthanization of rats for the biodistribution of minocycline at 3 h and targeted nanoparticle at 24 h were chosen because the half-life of minocycline is 2–3 h in rat models due to the extended systemic circulation of the targeted nanoparticle.

Minocycline extraction and Biodistribution analysis by HPLC: Prior to transcardial perfusion, rats were anesthetized with a combination of ketamine and xylazine (1:10 ratio) and perfused at either 3 h or 24 h post-blast. Blood (about 3 mL) was collected by cardiac puncture (left ventricle) and allowed to settle in vacutainer tubes (BD Bioscience) containing 3.2% sodium citrate for 10 min. Plasma was separated from blood by centrifuging at 2000 g. After the blood samples were collected, rats were transcardially perfused with PBS for isolation of the brain, liver, lungs, kidneys, heart and spleen. The tissues were homogenized, and the minocycline was extracted with sodium phosphate sulfite buffer and ethyl acetate using a liquid phase extraction [53].

Plasma aliquots (500 μL), tissue (homogenized) and standards (0.08 to 10.12 μM minocycline in plasma) were diluted with 1000 μL of sodium phosphate sulfite buffer (2.4 M disodium hydrogen phosphate, 4.0 M sodium sulfite, pH 6.5) and thoroughly mixed with 5 mL of ethyl acetate. After centrifugation to separate the phases, the aqueous phase was frozen, and the organic phase was poured off into 50 μL of 0.2% ascorbic acid and 0.1% cysteine in methanol. Samples were dried at 39 °C under nitrogen, and the residue was dissolved in 300 μL of HPLC mobile phase. After centrifugation, 12,000 rpm, 10 min, 50-μL aliquots were injected into the HPLC column and eluted with running buffer at 1 mL/min. Minocycline, with a retention time of 2.1 to 2.4 min, was detected at 270 nm. The minocycline concentration (per gram of tissue) in the supernatant was analyzed by using an optimized HPLC method [78,79].

#### Localization of Nanoparticle: Immunofluorescence Staining

Blast TBI rats were injected with fluorescently labeled targeted and non-targeted nanoparticles by tail vein. After 4 h, the brain was harvested and fixed with 4% paraformaldehyde for 24 h, incubated in a 30% sucrose solution for 72 h, and then stored at 80 °C until sectioning [80,81,82]. After blocking with 10% donkey serum for 1 h at room temperature, sections were incubated with microglia/macrophages (goat anti-Iba1, Invitrogen, 1:250 overnight at 4 °C), astrocytes (mouse anti-GFAP, Thermo Fischer Scientific, 1:400, overnight at 4 °C Primary antibodies were detected using donkey anti-goat and anti-mouse Alexa Fluor^®^ 488 (Invitogen, 1:1000, 1 h at RT). Coverslips were mounted using fluorescence anti-fade mounting media with DAPI.

### 2.8. Behavioral and Neuroprotection Study

#### 2.8.1. Novel Object Recognition (NOR) Test

Short-term object recognition memory was assessed in rats (*n* = 5/group) following moderate blast exposure using the NOR test at 2 and 36 days post-injury. Briefly, the NOR test consists of three phases: acclimation (day 1), familiarization (day 2), and testing (day 2). All phases were conducted in a 60 cm × 60 cm testing chamber. During the acclimation phase, each rat was allowed to explore the testing chamber for 5 min. During familiarization, rats were placed in the chamber with two identical objects and allowed to explore for 5 min before returning to their home cages. After 1 h, one of the identical objects was replaced with a novel object, and each rat was allowed to explore the testing chamber for 3 min. The total time spent exploring objects was recorded using ANY-Maze software (version 7.1, Stoelting, Wood Dale, IL, USA). A preference index [82] for the novel object was calculated as
$$\%\;preference\;index=\frac{\left(time\;spent\;with\;the\;novel\;object\right)}{time\;spent\;with\;both\;objects}\times 100$$

*Elevated Plus Maze (EPM):* This test was used to establish anxiety at acute and chronic time points such as 1 d and 35 d after moderate blast exposure. The elevated plus maze is a plus-shaped apparatus with four arms: two open and two closed arms. Each arm is 110 cm in length and the entire apparatus is elevated 60cm above the ground. Rats were placed at the center of the maze facing towards the open arm and given 10 min to explore the maze. Anxiety/depression-like behavior was determined by calculating the total time spent in the open arm in the maze. Animal movements in the maze were recorded by using the video camera positioned over the maze and the data was analyzed using the ANYMaze software [82].

*Open Field Test (OFT):* The open field test was utilized to identify gross-motor deficit, exploratory behavior and anxiety. The rat was left in an empty open field box set with a video camera (60 × 60 cm) for a period of 10 min. The rats were then tracked by the ANYMaze software and the total distance traveled by the rats, the time spent in the center of the field and time spent grooming time were measured for each group [76]. Immunohistochemistry *Neuron counting:* Neurons were manually counted (blinded) in ImageJ by identifying NeuN^+^ cells in CA1, CA3 and DG. Regions of interest (ROIs) were outlined using the polygon tool to define CA1, CA3 and DG in each image. Counts were normalized to each ROI area and reported as neurons per mm^2^ for each region. The granular cell layer of the dentate gyrus region was omitted from counting due to difficulties in quantifying densely packed neurons. 

#### 2.8.2. In Vivo Toxicity of Minocycline, Non-Targeted MANP and Targeted tfMANP

A drug delivery carrier should be able to deliver drugs to the target site effectively while ensuring efficacy [83]. The toxicity of minocycline and the nanoparticles in rats were analyzed by the following measures:

#### 2.8.3. Body Weight Examinations

For determining the toxicity of administered minocycline, non-targeted MANP and targeted tfMANP; gross observable body weight examinations, and histological assessments were performed [84]. Before treatment, body weight was measured for all the animals. For the observation period of 35 days, the animals were observed twice daily for indications of any adverse effects and were physically assessed for any sign of morbidity throughout the study period. The administration of minocycline, non-targeted MANP and targeted tfMANP was terminated after observing a single adverse behavioral change such as shivering, changes in awareness, motor activity and touch response. On days 0, 15 and 35, all the animal’s weights were measured.

#### 2.8.4. Gross Observable Behavioral Effects

The methods devised by Irwin (1968) were observed and the gross perceptible effects of minocycline, non-targeted MANP and targeted tfMANP were compared [84]. The animals were observed for a period of 35 days for their behavioral reactions. The animals under study were analyzed for adverse effect symptoms, such as changes in body weight, stool, condition of eyes and nose. Additionally, any general signs of distress such as writhing, color of skin and mucus membrane were also studied for any changes in the course of the treatment. 

#### 2.8.5. Hematoxylin and Eosin (H&E) Staining

Histological examination was adopted to demonstrate the toxicity of minocycline, non-targeted MANP and targeted tfMANP to the main organs. Rats were perfused with buffered (0.4 M phosphate buffer, pH 7.6) 4% paraformaldehyde. The brain, liver, lungs, kidneys and spleen were separated from each animal and immersion-fixed in the same fixative for 24 h at room temperature. The tissues were embedded in OCT, followed by cryostat sectioned. Hematoxylin and eosin (H&E) were used to stain the sections and the slides were observed by optical microscope to further investigate the potential signs of toxicity (i.e., cellular shrinkage, lesion or blebbing, steatosis in liver cells, condensation of chromatin, rupture of the cell membrane and apoptotic bodies) [83,84,85,86,87]. 

### 2.9. Statistical Analysis

Data are given as mean ± standard error of the mean (SEM). Between-group comparisons were made by one-way and two-way analysis of variance (ANOVA) mixed model design with a post hoc test (Tukey) to determine individual group differences. Differences between means were evaluated at the probability level of *p* ≤ 0.05, 0.01, and 0.001. GraphPad Prism 6.0 software was employed in all analyses and preparation of plots.

## 3. Results and Discussion

Minocycline-loaded albumin nanoparticles were lyophilized with 0.05% mannitol, leading to a brownish powder that can also be dispersed in PBS or 0.9% saline solution (Figure 2a,b). Nanoparticles of identical size permit increased cellular interaction and have increased toxicity [88]; hence, the size of the nanoparticle plays a crucial role in cellular interactions and toxicity. The particle size distribution and mean particle size of nanoparticles were measured by DLS in PBS at pH 7.4. The results show a well-dispersed colloidal system of MANP nanoparticles with a mean particle size and polydispersity index of 135.4 ± 5 nm and 0.275 ± 0.01, respectively, and with a distribution range of 140–250 nm (Figure 2c). In the case of ligand-conjugated minocycline loaded nanoparticles (tfMANP), a mean particle size and polydispersity index of 153.5 ± 3.4 nm and 0.203 ± 0.5 nm, respectively, were observed (Figure 2d). Studies have reported that cellular uptake and cytotoxicity depend on the surface charge of the nanoparticle [89]. Lyophilization may increase the particle size of a nanoparticle, conceivably due to aggregation; consequently, the redispersibility of the particles was checked following lyophilization using mechanical shaking and bath sonication methods. 

Morphological images of freeze-dried MANP and tfMANP samples was were captured with FE-SEM, and the images obtained are shown in Figure 2e,f. Both types of synthesized NPs had an approximately spherical morphology and were to some extent, connected to each other with an average size of 120 ± 30 nm for MANP and 148 ± 6 nm for tfMANP synthesized at pH 7. Furthermore, tfMANP showed a majorly uniform distribution, minimally differing in size when compared to MANP. TEM analysis showed that MANP and tfMANP samples have a spherical shape and are similar in size range, measuring 120–180 nm in diameter (Figure 2g,h) when compared to each other with SEM and zetasizer analysis. Zeta potential measures the amount of repulsive interaction among nanoparticles and depends on the concentration of the polymer and the incorporated drug. Specifically, zeta potential is calculated as the difference in electrical potential between the surface of the nanoparticle and the bulk-surrounding medium [90]. Zeta potential information is helpful in predicting the storage stability of colloidal dispersions [64]. In our study, MANP and tfMANP exhibited an anionic charge of −2.52 ± 1.2 mV and −3.14 ± 3.4 mV, respectively. Albumin is comprised of free carboxyl and amine groups, both of which can be applied for covalent modification. Overall, albumin shows a negative zeta potential (high anionic) at physiological pH and saline. Furthermore, negative albumin (nearly neutral) can also be obtained with surface modifications via anionic groups [44].

### 3.1. Minocycline-Loaded Albumin Nanoparticles (MANP)

Our formulation of minocycline-loaded albumin nanoparticles (MANP) is based on a modified desolvation method [57]. Many studies have reported albumin-based drug-loaded nanocarriers with chemical crosslinking agents such as glutaraldehyde [91,92]. In our study, parameters such as pH level (between 7.0 and 9.0) and the amount of ethanol added were varied to optimize particle size (Appendix A). Though the initial particle size was on the higher end of the scale (>275 nm), by the end of the ethanol addition the particle size stabilized. Size became reproducible following glutaraldehyde cross-linking, rotary evaporation of ethanol, and finally, lyophilization. Moreover, at 10% *w*/*v* BSA and a 1:1.3 addition of BSA to ethanol, we obtained a maximum yield of 68.9%. Particle sizes were higher when pH was <7.0; however, pH levels between 7.0 and 9.0 produced uniformly sized albumin nanoparticles. When the ratio of BSA to ethanol was 1:2.5, the yield was higher than the 1:5 ratio. Subsequently, at the 1:2.5 ratio of BSA to ethanol, the particle size also increased to 320 nm prior to ethanol removal. Upon ethanol removal and lyophilization, the particle size was reduced to the optimal size and the PDI indicated a nearly homogenized nanoparticle system. 

FTIR spectra of BSA, Transferrin, minocycline, MANP and tfMANP was obtained (Figure 3). The FTIR spectra of BSA displayed distinguishing peaks at 3319 cm^–1^ arising from the amine groups (N-H stretching vibration, a peak of amide bond at 1659 cm^–1^ attributed to the C=O stretching vibration (amide I band), and a mixed vibration of N-H bending and C-N stretching (amide II band) at 1533 cm^–1^ [58]. Minocycline exhibited characteristic bands at 3487 cm^−1^. The additional bands at 1597 cm^−1^ and 1473 cm^−1^ are due to the structural vibrations of the benzene rings [93]. The absorption band at 1042 cm^−1^ was due to C–O stretching vibrations. MANP and tfMANP showed characteristic bands at 3466 cm^−1^ and 1653 cm^−1^ related to the O–H alcohol and C=O groups of acids [94]. These results show the absence of a chemical reaction between BSA and minocycline, suggesting that nanoparticle formation does not modify the chemical structure of the drug. The presence of relevant peaks in each nanoparticle confirmed that minocycline was encapsulated in the BSA nanoparticles.

The BSA, Transferrin, minocycline, and tfMANP conjugates were then characterized by proton nuclear magnetic resonance spectroscopy (1H-NMR) to confirm successful synthesis. The 1H-NMR analyses were logged on a Bruker DRX-600 Avance III spectrometer with room-temperature deuterated chloroform (CDCl3) as the solvent. DHA, 8arm-PEG-DHA and TF-8arm-PEG-DHA (1 mg of each) were dissolved in 1 mL of CDCl3, and the products were evaluated using an NMR spectrometer. The results showed that there is no specific peak of minocycline in the nanoparticles which confirmed the lack of chemical interaction and provided evidence for a physical mixture of the nanoparticle and BSA (Figure 4a). However, the peak at 3.5 confirmed that the ligand is conjugated to the nanoparticle and thereby corroborates the FTIR results.

### 3.2. Entrapment Efficiency and In Vitro Release Study

In order to develop an optimal drug delivery system, encapsulation efficiency and in vitro drug release profiles are two essential parameters that must be analyzed [95]. Our formulation of minocycline-loaded albumin nanoparticles is the first to report these results as there have not been any chromatographic methods to quantify minocycline from albumin nanoparticle formulations. In this study, we used the HPLC method for the quantification of minocycline from a novel albumin nanoparticle. The encapsulation efficiency of minocycline in the albumin nanoparticles was determined indirectly by centrifuging and collecting the supernatant of the nanoparticles and estimating the free minocycline present in the solution [72]. We determined nearly 59.4 ± 1.7% and 0.45 ± 0.06% of entrapment efficiency and drug loading rate, respectively (*n* = 3). The in vitro release profile of minocycline and minocycline-loaded nanoparticles is shown in Figure 5. An initial burst release of minocycline was observed after 30 min (22.9 ± 1.3%) in dissolution medium, when compared to free minocycline (61.5 ± 2.4%), which gradually increased until 24 h. This indicated that the surface-implanted minocycline is released earlier than inside the nanoparticle. At 24 h, approximately 70.3 ± 2.9% of the minocycline was released compared to free minocycline (98.3 ± 1.8%). The release rate gradually stabilized to a sustained from 24 h to 72 h, which can be attributed to minocycline’s soluble nature and the formation of interlinked pores between the surface and insides of the albumin nanocarrier upon contact with the dissolution medium. Furthermore, NPs with glutaraldehyde crosslinking on the surface amino groups displayed slower kinetics of drug release [58].

### 3.3. Stability Study

The size and polydispersity index were measured over a 3-month period using lyophilized minocycline loaded nanoparticles stored at two different conditions (4 °C and 25 °C). The stability of the nanoparticle is shown in Figure 6. 

Freeze-dried nanoparticles stored at 25 °C displayed a considerable particle size increase. With the particle size attaining approximately 269.2 ± 11.2 nm within 2 months, this system would be considered undesirable for use as a brain-targeted drug delivery system. Minocycline-loaded nanoparticles stored at 4 °C were stable and displayed greater uniformity in their particle size measurements during all time periods. For PDI values, nanoparticles stored at both 25 °C and 4 °C conditions exhibited a value of <0.3, in accordance with the acceptable limit of monodispersed nanoparticle distribution [72,96,97]. Nonetheless, increased PDI was also observed over the 2-month time period under all conditions except the lyophilized powder stored at 4 °C. Therefore, lyophilized nanoparticles stored at 4 °C revealed superior stability as compared to those stored at 25 °C conceivably due to aggregation during longer storage. Therefore, to ensure nanosized uniform particles for subsequent studies, our formulations were freshly prepared as lyophilized powder, stored at 4 °C, and avoided precipitation by continuous mixing preceding use.

### 3.4. Enhanced Biodistribution of Nanoparticle/Minocycline in bTBI Rat Model

The bioavailability to the organ of interest is an important criterion since low bioavailability requires the use of excessive dosage, which results in triggering serious side effects in the central and peripheral nervous systems. We have performed in vivo biodistribution studies of minocycline using targeted nanoparticles in a rat model of TBI. The results show maximal concentration (~3′ fold higher) of minocycline accumulation in the brain within the group administered targeted nanoparticle (after 24 h) as compared to the free minocycline injection group. This is due to the transferrin (tfr) receptor (BBB)-mediated uptake of the tf conjugated nanoparticle in brain tissue as well as the short half-life (2–3 h) of free minocycline. The bioavailability of higher concentrations of minocycline administered for longer periods of time (24 h) in the brain at single doses of a nontoxic concentration (3 mg/kg) of targeted nanoparticle will be compared to free minocycline administration and will serve as a proof-of-concept. 

Both free minocycline and tf conjugated MANP were compared in terms of their biodistribution in different organs using a rat model of TBI at 4 and 24 h post-administration. Our study demonstrated that administering the targeted nanoformulation achieved 3-fold more bioavailability as compared to administering free minocycline (Figure 7). Meanwhile, other organs such as the spleen and lungs displayed lower minocycline (nanoparticle) distribution as compared to the brain and free minocycline group. This enhanced delivery of minocycline to the brain compared to other organs can be attributed to particle size, surface charge, and transferrin. These factors assisted in increasing the permeability of the nanoparticle into the brain and subsequent interaction with the cell membrane. It is also possible that the compromised BBB in blast-induced brain injury may have also contributed to the enhanced delivery of minocycline and the targeted MANP. We postulate that the P-gp efflux pump located at the endothelial cells of the BBB plays an important role in the efflux mechanism of the brain [53]. In these respects, minocycline may be effluxing out of the brain. In the case of our targeted nano construct MANP, tf may be being recognized as an endogenous unit and thus enabled it to achieve higher concentrations inside the brain for a relatively prolonged period of time. Interestingly, we also observed higher distributions of the nanoparticles in the liver. This can be explained as the endothelial barrier determines distribution into organs [98]. Macromolecules and nanoparticles extravasate primarily by convection, with its relative contribution increasing with transcapillary volume flow. The extent of crossing the endothelial barrier is dependent on the capillary pore size, which generally forms the upper particle size limit [98]. Depending on the organ or tissue, the endothelial gap junctions differ from very tight (blood–brain barrier) to continuously small (standard arteries and capillaries) and fenestrated (digestive mucosa). The liver, as well as lymphoid tissue and hematopoietic organs, comprises sinusoidal (discontinuous) capillaries. Unlike fenestrations, these capillaries lack a diaphragm over the pore, and thus show even superior permeability, allowing the exchange of macromolecules. As mentioned before, siRNA delivery systems have an inherent preference for the liver. This pertains to lipid-based particles, cationic and neutral nanoparticles, and lipophilic ligands in bioconjugates. Liver accumulation is caused by the discontinuous nature of the hepatic vasculature (for nanoparticles and protein-bound oligonucleotides) or the lipid metabolism (liposomes and lipid or cholesterol conjugates). Within the liver, the use of a specific receptor-targeted ligand can be useful to distinguish between the different hepatic cells and prevent an accumulation in Kupffer cells, the resident macrophages in this organ. Thus, trivalent GalNAc conjugates increase uptake into hepatocytes and consequently gene silencing in this cell type [98]. The asialoglycoprotein receptor is characterized by efficient endocytosis. Nanoparticles with close to neutral and anionic charges have been described to decrease the adsorption of serum proteins, leading to increased circulation half-lives [99]. Kataoka et al. reported that neutral and anionic nanoparticles tend to circulate for longer periods of time and subsequently lead to lower accumulation in the spleen and other organs [100]. Meanwhile, cationic nanoparticles have been shown to have increased cellular localization in most of the cells as shown in Figure 8 [101]. 

### 3.5. Behavior, Neuroprotection and Toxicity Analysis of Minocycline and Its Nano-Formulations

#### Free, Targeted and Non-Targeted Minocycline Protected against Moderate Blast-Induced Chronic Memory Impairments

To assess the effects of free minocycline, targeted (TMNP), and non-targeted (NTMP) minocycline nanoparticles on short-term object recognition memory after moderate blast exposure, rats (*n* = 5/group) underwent a novel object recognition (NOR) task at 2 days and 36 days post-injury. The preference for the novel object significantly decreased in the blast group at 36 days (*p* < 0.05) post-injury compared to controls such that injured rats spent more time with the familiar object than the novel object as a result of short-term memory deficits (Figure 9). Whereas, free minocycline, TMNP, and NTMP-administered groups showed more preference compared to the blast group. Additionally, the TMNP groups showed more preference for the novel object compared to blast, free, or NTMP minocycline groups (*p* < 0.0001) which implies TMNP protects from chronic memory impairments. We also observed variability in behavioral improvements from day 2 to day 36 between the groups. Free minocycline showed a profound impact on day 1 on the preference index but the impact reduced dramatically by day 36 which implies clearance of free minocycline from the body. TMNP and NTMP did not show an impact greater than free minocycline but unlike free minocycline, by day 36, they still showed a similar impact as in day 1, especially TMNP due to the sustained release of the minocycline drug from the nanoparticle and slower degradation from the body. Therefore, TMNPs sustain in the brain for longer periods of time, manifest neuroprotection, and improve memory impairments. However, we did not see any difference between the groups at the acute time point.

### 3.6. Free, Targeted and Non-Targeted Minocycline Reduced Moderate Blast-Induced Chronic Anxiety

To assess the effects of free minocycline, targeted, and non-targeted minocycline nanoparticles on acute and chronic anxiety after moderate blast exposure, rats (*n* = 5/group) underwent an elevated plus maze (EPM) task at 1 day and 35 days post-injury (Figure 10). Like NOR, the blast group spent less time in the open arm compared to all other groups on days 1 and 35 post-injury. However, all these groups did not show any significance at any time point however a similar trend as the NOR task was observed. Additionally, the targeted minocycline nanoparticle group spent more time in the open arm compared to blast, free, or non-targeted minocycline groups. Hence, a single administration of free, targeted, and non-targeted nanoparticles showed neuroprotective roles and improved behavioral changes. 

### 3.7. Free, Targeted and Non-Targeted Minocycline Reduced Moderate Blast Induced Gross Motor Movements, Exploratory Skills and Chronic Anxiety

To assess the effects of free minocycline, targeted, and non-targeted minocycline nanoparticles on the acute and chronic effect of anxiety level and motor function changes and, exploratory behaviors after moderate blast exposure, rats (*n* = 5/group) underwent an open field test (OFT) task at 1 day and 35 days post-injury. The blast group spent less time in the center zone and groomed more as compared to all other groups on days 1 and 35 post-injury. Throughout the behavioral tasks, all groups did not show any significance at any time point; however, a strong trend was observed in all tasks. TMNP rats spent more time in the center zone and performed less grooming compared to blast, free, or non-targeted minocycline groups. Overall, moderate blast-induced chronic memory impairments, anxiety, and gross motor impairments (Figure 11). TMNP persistently improved behavioral outcomes at chronic time points compared to free and NTMP minocycline nanoparticles. The therapeutic effect of TMNP can be attributed to sustained release and lesser clearance from the body.

Targeted and non-targeted minocycline showed neuroprotection against blast-induced neuronal loss.

Since we observed significant short-term memory improvements in NOR task, we examined whether free minocycline, TMNP and NTMNP had a neuroprotective role against neuronal loss following moderate blast TBI. We quantified NeuN+ cells in CA1, CA3 and DG regions of the hippocampus using immunohistochemistry (as shown in the Appendix A). Compared to free minocycline, TMNP and NTMNP showed more neuroprotection against blast-induced neuronal loss in the CA3 and DG regions. Additionally, TMNP-injected rats showed neuroprotection in the CA1 region as well (Figure 12). This could be due to the slower degradation and sustained release of minocycline resulting in increased neuroprotection in blast TBI animals. Studies have demonstrated minocycline’s protective role against neuronal cell death in various disease models [102]. This protection could be due to mitigated glial cell activation (microglia and astrocytes) mediated neurotoxicity [103,104]. Furthermore, in Figure 8, we showed that the nanoparticles delivered to the parenchyma reached glial cells where minocycline could have suppressed effects on glial activation. Earlier studies have reported that glial activation and neuroinflammation can result in memory impairments [105,106]. In our previous work, we also showed that moderate blasts can cause acute and chronic glial activation and short-term memory impairments [76]. In particular, the TMNP group has persistently shown higher levels of neuroprotective effects compared to free minocycline and NTMNP in behavioral tasks and biochemical studies. Here, we have shown that transferrin-mediated nanoparticle delivery certainly has long-term therapeutic effects. 

### 3.8. Toxicity of Minocycline and Nanoparticle

A comparison of transverse sections of the liver, lungs, kidneys and spleen revealed no abnormal histopathological changes or lesions in the minocycline, non-targeted MANP or targeted tfMANP treated groups (Figure 13a). Additionally, there was no neutrophil or monocyte/macrophage infiltration in the liver, kidney, lung or spleen (Figure 13a). H&E staining demonstrates no architectural changes in the liver, kidney, lung or spleen following injection of the minocycline, non-targeted MANP or targeted tfMANP (Figure 13a). However, certain areas showed the irreversible condensation of chromatin in the nucleus of a cell undergoing pyknosis. Nuclear chromatin that appeared dark when stained revealed evidence of hyperchromasia. Mild or negligible liver pathologies relating to minocycline, non-targeted MANP or targeted tfMANP injection included clusters of infiltrating cells that were sporadically found in the liver, and few necrotic hepatic cells [85]. However, none of these changes were found to be significant. 

Others have also reported that the formulation of minocycline nanoparticles resulted in decreased accumulation of the drug in the liver, spleen, kidney and heart [84]. Because of the nanoparticle formulation of minocycline, we observed decreased accumulation of the drug in the liver, spleen, kidney and heart, indicating reduced toxic effects of minocycline in these tissues [84]. The minocycline nanoparticle concentration was unable to produce any adverse effects on brain tissue (Figure 13a). Reduced off-target uptake of the PEGylated nanoparticles may reduce the toxicity of the drug. Our results are inconsistent with these previous findings. Since these nanoparticles were designed to be targeted for delivery to brain endothelial cells, the treatment of non-targeted MANP and targeted tfMANP considerably reduced the hematological adverse effects of pure minocycline. 

Body weight examinations: Neither the free drug nor the nanoformulation should induce overt toxicity nor greater than 10% retardation of body weight as compared to control animals following injection [84]. Baseline body weight measurements were determined for all animals. As compared to the saline-treated control group, treatment with minocycline, non-targeted MANP or targeted tfMANP injection showed no reduction in body weight of the Sprague-Dawley. Conversely, all the treatment groups showed increased body weights similar to the control groups (Figure 13b). Taken together, repeated administration of minocycline, non-targeted MANP or targeted tfMANP did not cause any apparent toxicity [86]. Furthermore, no significant toxic effects were observed in any of the injected groups. 

## 4. Conclusions

Minocycline’s neuroprotective role can be attributable to its anti-inflammatory property; however, in order to see any therapeutic benefits, minocycline must be administered at higher or multiple dosages even though minocycline can cross the BBB. Administering multiple or higher doses has been shown to cause severe neurotoxicity or side effects which have halted the wide usage of minocycline as a treatment for neurological diseases. In order to use minocycline as a neuroprotective therapy, there is a need for developing novel targeted delivery approaches to the brain to minimize dosage and multiple administrations. As a preliminary approach, to increase drug availability in brain tissues, we have attempted to optimize the formulation parameters of minocycline-loaded albumin nanoparticles prepared through the desolvation method. Our study indicates that the formulation we have developed produces monodispersed, nano-sized particles with higher entrapment efficiency, loading rate, stability, and sustained release profiles. We have developed a precise, accurate, specific and stable HPLC method for quantifying minocycline. In this work, we successfully synthesized minocycline-loaded albumin nanoparticles and showed improvement in cognitive deficits as well as long-term neuroprotective effects with a single dose administration in blast TBI. tf-MANP was successful in delivering relatively high amounts of minocycline to the brains compared to free minocycline administration and resulted in improved biodistribution 24 h post-drug administration. We conclude that tf-conjugated MANP can be used as a non-toxic, efficient and specific ligand for brain-targeted drug delivery that achieves a relatively high minocycline concentration in the brain. Therefore, the current study significantly complements the research and development efforts of novel systemic targeted drug delivery systems containing minocycline in addition to other therapeutic molecules.

## 5. Patents

Non-provisional application filed on 27 April 2022. V. Perumal, AR. Ravula, N. Chandra Targeted Nanoparticle for Traumatic Brain Injury and other CNS Diseases, U.S. Serial No. 17/730,358.

## Figures and Tables

**Figure 1 brainsci-13-00402-f001:**
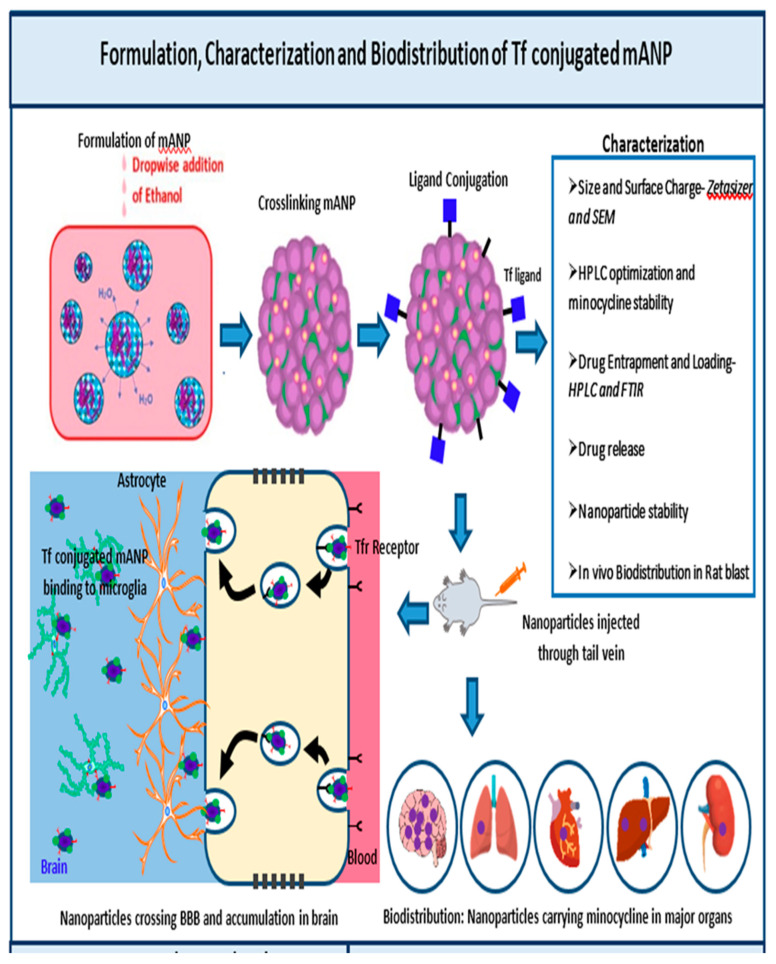
Scheme of the formulation of minocycline loaded transferrin conjugated albumin nanoparticle (tf-mANP) including characterization, biodistribution analysis and efficacy study in a rat model of bTBI.

**Figure 2 brainsci-13-00402-f002:**
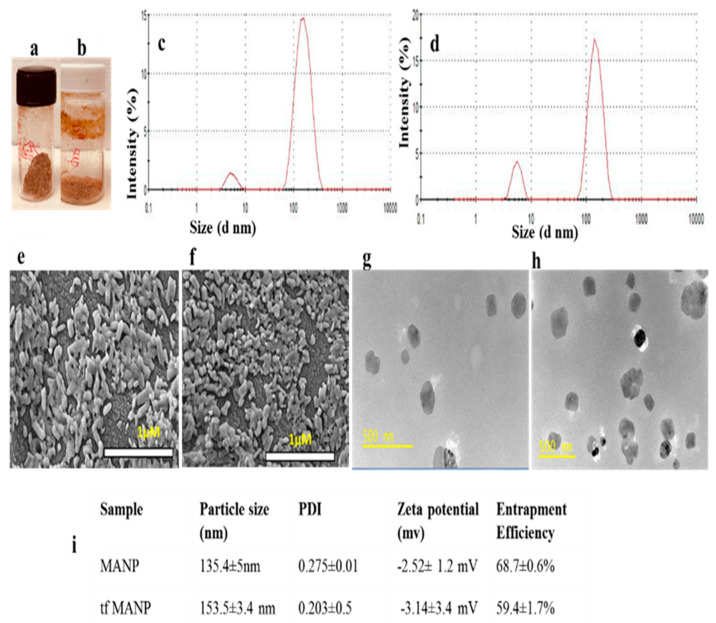
Characterization of the physical and chemical properties of MANP and tfMANP. Freeze-dried sample of (**a**) MANP and (**b**) tfMANP; Particle size of (**c**) MANP and (**d**) tfMANP; Representative SEM images for (**e**) MANP (magnification =25,000×) and (**f**) tfMANP (magnification = 30,000×). Representative TEM images for (**g**) MANP (magnification = 26,000×) and (**h**) tfMANP (magnification = 26,000×). (**i**) Particle size, PDI, zeta potential and Entrapment Efficiency Data expressed as mean ± SD (*n* = 3), statistical significance (*p* < 0.05), PDI: Polydispersity Index.

**Figure 3 brainsci-13-00402-f003:**
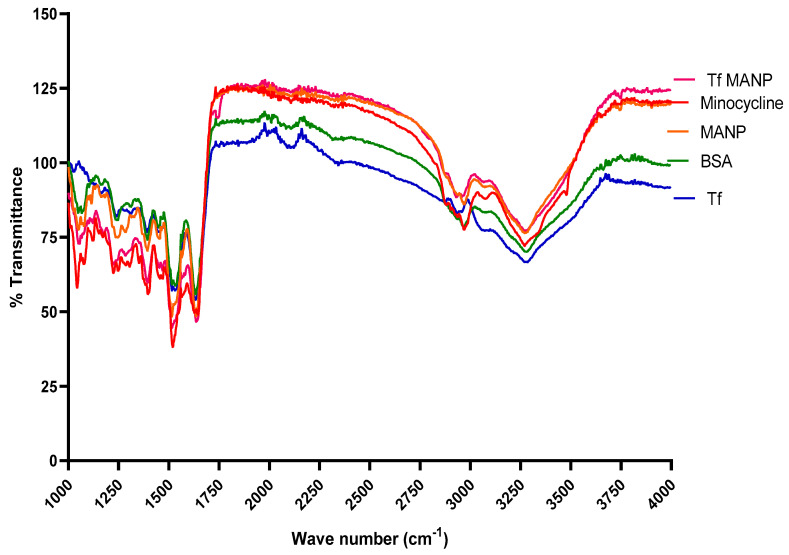
FTIR spectra of BSA; Minocycline; Transferrin; MANP; and tfMANP.

**Figure 4 brainsci-13-00402-f004:**
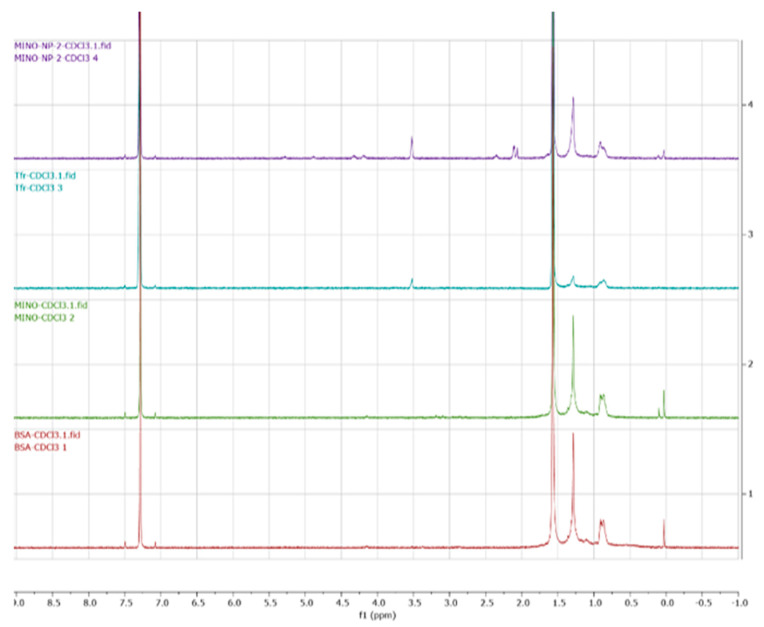
NMR spectra of BSA, Transferrin, minocycline, and tfMANP conjugates.

**Figure 5 brainsci-13-00402-f005:**
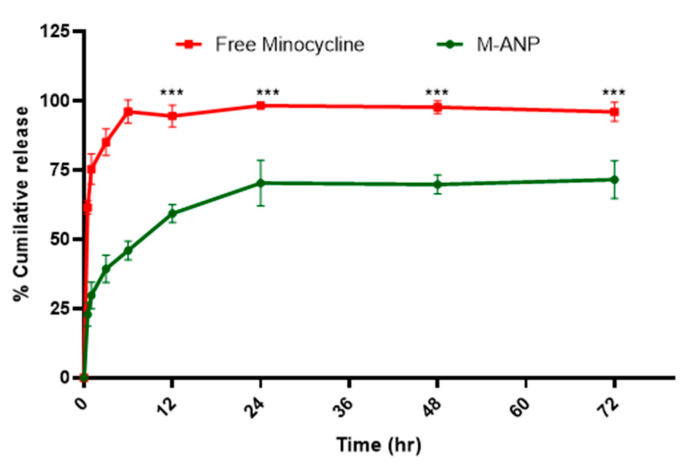
Drug release profile of minocycline loaded Albumin Nanoparticle. Values shown here represent SD (*n* = 3). Two-way Anova, Multiple comparison analysis *** *p* < 0.001.

**Figure 6 brainsci-13-00402-f006:**
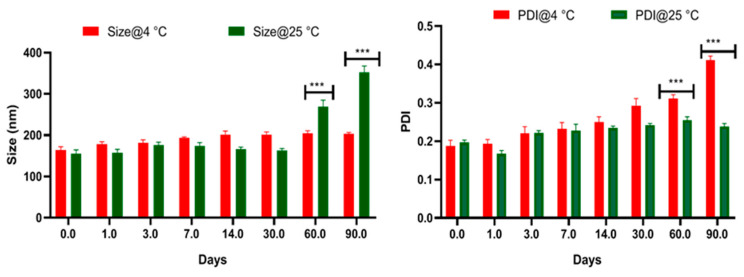
Stability analysis of minocycline loaded albumin nanoparticle performed by measuring change in particle size and PDI of lyophilized powder stored at either 4 °C or 25 °C over a 3-month time. Values shown here represent SD (*n* = 3). Two-way Anova, Multiple comparison analysis *** *p* < 0.001.

**Figure 7 brainsci-13-00402-f007:**
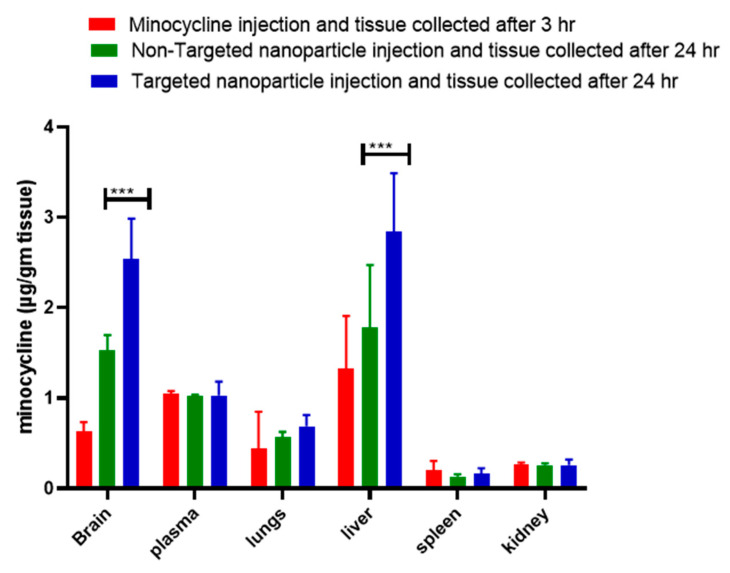
Biodistribution of minocycline and targeted nanoparticle in a rat model moderate blast TBI. Two-way Anova, Multiple comparison analysis *** *p* < 0.001.

**Figure 8 brainsci-13-00402-f008:**
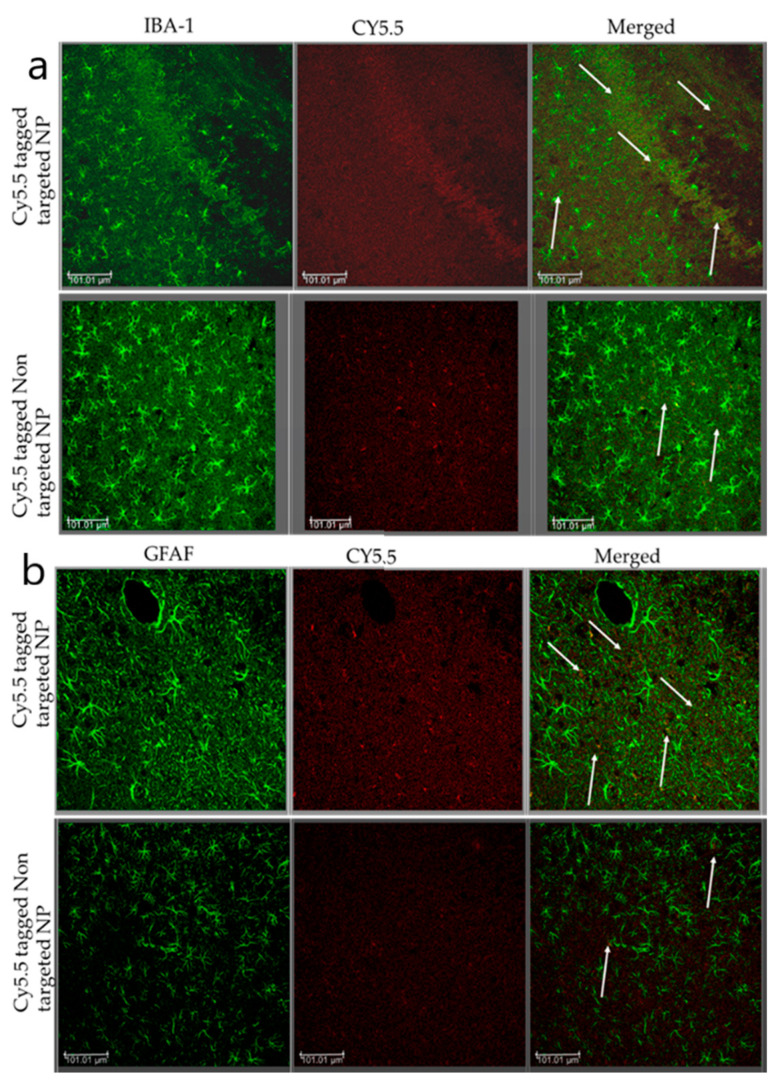
Nanoparticles internalized in the brain parenchyma and glial cells by crossing the BBB and actively targeting via transferrin receptors. bTBI rats were intravenously injected with Cy5.5 conjugated MANP or tfMANP and then sacrificed 6 h after the injection. Brain tissues were collected 6 h after injections of Cy5.5 conjugated MANP or tfMANP. Sections were observed under a confocal microscope. The distribution in the brain parenchyma and uptake of Cy5.5 conjugated MANP and tfMANP (white) by (**a**) microglia and (**b**) astrocytes is shown. Scale bar = 100 μm.

**Figure 9 brainsci-13-00402-f009:**
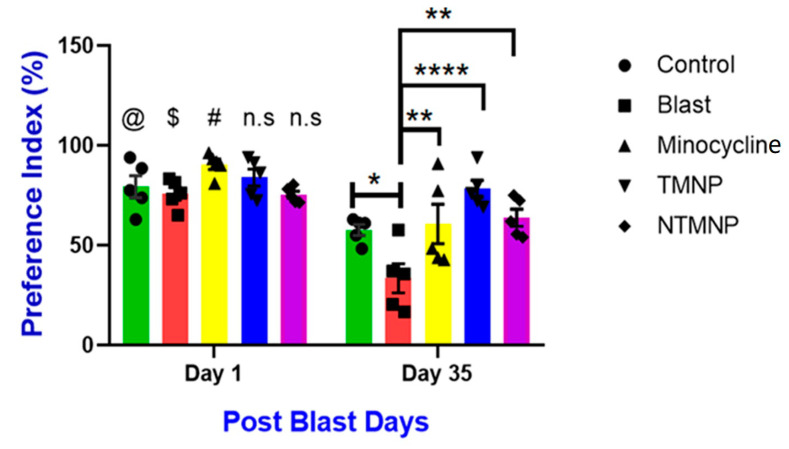
Free minocycline, targeted (TMNP), and non-targeted minocycline (NTMNP) nanoparticles mitigated moderate blast-induced short-term memory impairment. @, $ and # represent significant differences between day 1 and day 35 within the groups. Two-way ANOVA-mixed effect model design with Bonferroni correction used to analyze the difference between the days). * *p* < 0.05, ** *p* < 0.01, **** *p* < 0.001, and n.s is no significance.

**Figure 10 brainsci-13-00402-f010:**
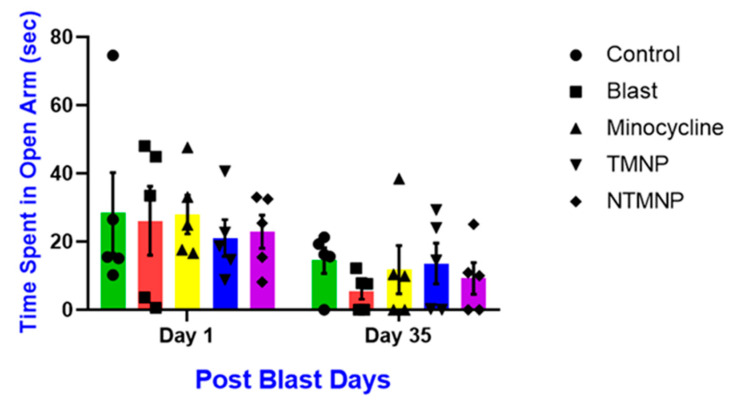
Free minocycline, targeted (TMNP), and non-targeted minocycline (NTMNP) nanoparticles reduced moderate blast-induced anxiety.

**Figure 11 brainsci-13-00402-f011:**
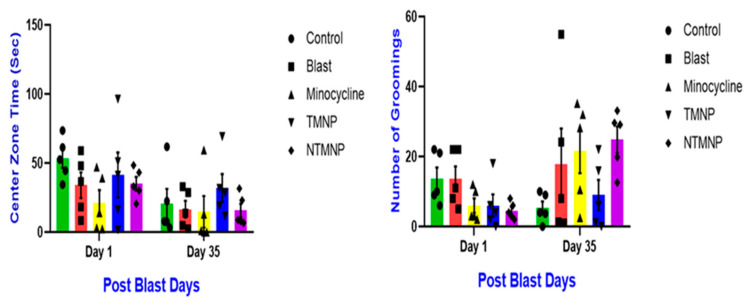
Free minocycline, targeted (TMNP), and non-targeted minocycline (NTMNP) nanoparticles reduced moderate blast-induced gross motor skills and anxiety.

**Figure 12 brainsci-13-00402-f012:**
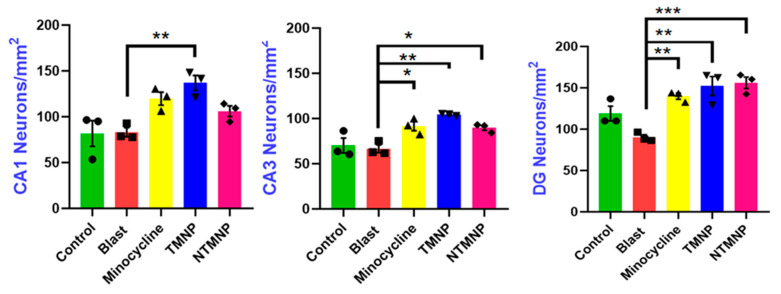
Free minocycline, targeted (TMNP), and non-targeted minocycline (NTMNP) nanoparticles showed neuroprotection against blast-induced neuronal cell death (*n* = 3). * *p* < 0.05, ** *p* < 0.01, *** *p* < 0.001.

**Figure 13 brainsci-13-00402-f013:**
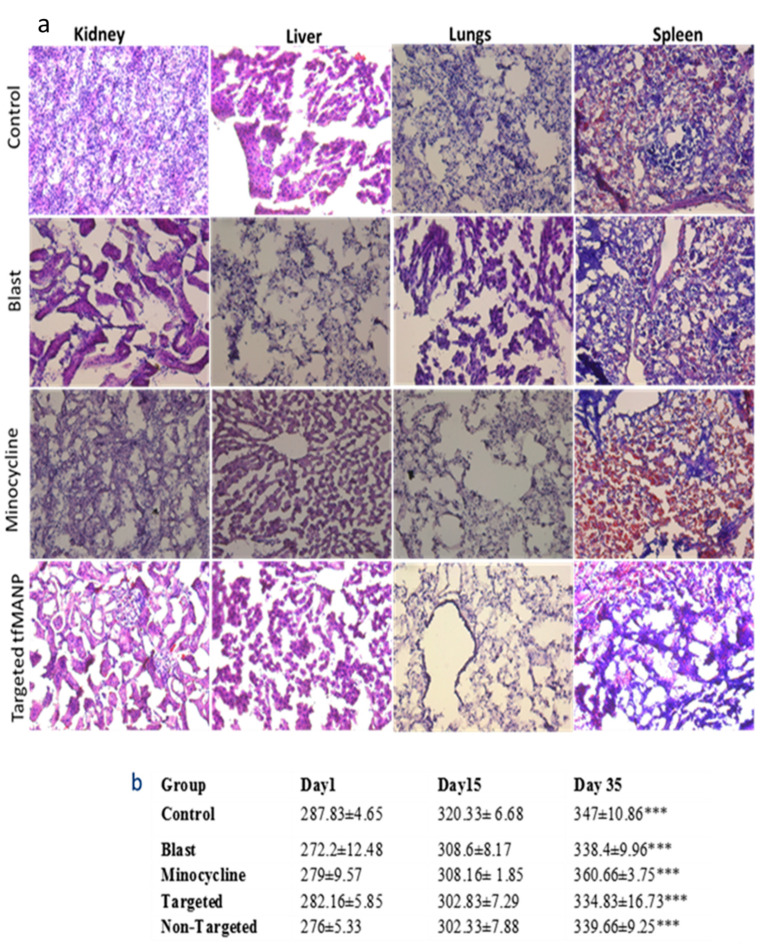
(**a**) Histopathological evaluation of the major organs of rats treated with physiological saline, minocycline, non-targeted MANP (non-targeted) or targeted tfMANP (targeted). Histological analysis of the organs (liver, kidneys, lungs and spleen) compared to the control group. No abnormal histopathological findings were observed in the liver, kidneys, lungs or spleen; (**b**) Body weight changes in blast TBI rats treated with physiological saline, blast, minocycline, non-targeted MANP (non-targeted) or targeted tfMANP (targeted). *** *p* < 0.001.

## Data Availability

Data is unavailable due to privacy or ethical restrictions.

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
