# Peer review of "Enhanced Targeted Delivery of Minocycline via Transferrin Conjugated Albumin Nanoparticle Improves Neuroprotection in a Blast Traumatic Brain Injury Model"

_brainsci, 2023, doi:10.3390/brainsci13030402_

Round 1

Reviewer 1 Report

the study conducted by Perumal et al is interesting but it has many issues.

1- in fig 8, there is no DAPI staining. authors should also need to add negative control (secondry antibodies only).

2- Poor statistical analysis

A-(in Fig 9 please use two way annova and do statistical analysis between day 1 and day 35 also

b-fig 12- all the analysis was done with blast, please compare each group with control also

3- fig 12- most of the images have diff background. pleas provide all the raw images in supplementary to make sure that image were taken at same time and they have appropriate number of n.

4- please validate your findings by western blot also and use molecular weight  markers and full blot also.

Author Response

We sincerely thank you for your comments on our manuscript entitled “Enhanced Targeted delivery of minocycline via transferrin conjugated Albumin Nanoparticle and improves neuroprotection in a Blast Traumatic Brain Injury Model”. These comments are all valuable and helpful for us to revise and to improve our manuscript. We have evaluated the comments carefully and have made the necessary corrections. We hope this will meet with your approval. Revised portions are marked in the manuscript as track change and the main corrections and the response to reviewer’s comments are listed below.

We sincerely thank the reviewer for acknowledging our research work. Yes, we have made necessary corrections throughout the manuscript for English language corrections using 3 different experts in English writings.

The study conducted by Perumal et al is interesting but it has many issues.

We sincerely thank the reviewer for acknowledging our research work. Yes, we have made necessary corrections to address reviewer comments.

  1. 1- in fig 8, there is no DAPI staining. authors should also need to add negative control (secondry antibodies only).

Thanks for reviewer suggestion. The study was mainly focused on localization of nanoparticles in glial cells like microglia and GFAF, and brain parenchyma. We were concerned about the overlapping of intensity of CY5.5 with DAPI. Hence, we standardized the protocol without DPAI staining.

  1. Poor statistical analysis

A-(in Fig 9 please use two way annova and do statistical analysis between day 1 and day 35 also

 Thanks for reviewer suggestion. We have mentioned in statistics section, we used two-way Annova for analyzing behavioral outcome with Tukey post hoc test. Further we also included the Statistical analyses day 1 and day 35 comparison as suggested by reviewer and the results are included in the manuscript.

b-fig 12- all the analysis was done with blast, please compare each group with control also.

We apologies for missing out the comparison with control. We did observe a significant difference between control vs treatment groups but not in blast injury group. Hence, we did not mention it in the manuscript.

 - fig 12- most of the images have diff background. pleas provide all the raw images in supplementary to make sure that image were taken at same time and they have appropriate number of n.

We have included the few representative images (Fig. S8) in the supplementary file as requested by the reviewer.

  1. - please validate your findings by western blot also and use molecular weight  markers and full blot also.

 Thanks for the reviewer suggestion. As this is a preliminary study, we performed only IHC for the biochemical analysis. Future studies are planned to perform detailed molecular pathway analysis including western blot.

In terms of English correction and plagiarism Ithenticate plagiarism “Some of the technical (material and methods) Even though we have already cited reference articles, plagiarism software is identifying the whole material and methods sections as plagiarism that we are unable to revise or rephrase. Ithenticate plagiarism software also picks up (that amount to total 32%, revised manuscript should be very less) even individual words in a sentence, and subheadings and fig legends that we are unable rephrase or unable to avoid plagiarism. We would like to give the article a brief overview of “materials and methods” and cited appropriate articles for readers understanding. If you are still concerned, we can cut down whole just cite our previously published article with our details. Hope you will be able to understand.

Further, Scientific terminology (for ex., intravenous inj, neurological disease names like as Alzheimer’s disease, Parkinson’s disease, Huntington’s disease, focal cerebral ischemia, multiple sclerosis, traumatic brain injury and spinal cord injury) and the nouns like liver, spleen kidney and body weight were identified by ithenticate plagiarism software.

Following are some of the examples of Plagiarism identified by ithenticate  that we are unable to rephrase or to avoid.

  • “Animals: Ten-week-old male Sprague-Dawley (Charles River Laboratories) rats at 350± 50 g were used in accordance with protocols approved by Rutgers University In-stitutional Animal Care and Use Committee (IACUC approval: PROTO201900142). The animals were housed with free access to food and water in a 12-h dark-light cycle at 22 °C. Rats were divided into five groups (n=6); sham controls, blast and three treatment groups of animals exposed to a moderate blast of 180 kPa)[18,74].”

  • “Rats were perfused with buffered (0.4 M phosphate buffer, pH 7.6) 4% paraformaldehyde. The brain, liver, lungs, kidneys, and spleen, were removed from each animal and immersion-fixed in the same fixative for 24 h at room temperature”.

  • “Statistical analysis

Data are presented as mean ± standard error of the mean (SEM). Between-group comparisons were made by one-way and Two-way analysis of variance (ANOVA) mixed model design with a post hoc test (Tukey) to determine individual group differences. Differences between means were assessed at the probability level of p ≤0.05, 0.01, and 0.001. GraphPad Prism 6.0 software was used in all analyses and preparation of plots”.

  • “Minocycline exhibited characteristic bands at 3487 cm−1. The additional bands at 1597 and 1473 cm−1 are due to the structural vibrations of the benzene rings[97]. The absorption band at 1042 cm−1 was due to C–O stretching vibrations. MANP and tfMANP showed characteristic bands at 3466 and 1653 cm−1 related to the O–H alcohol and C=O groups of acids[98].”

  • “The BSA, Transferrin, minocycline, and tfMANP conjugates were then characterized by proton nuclear magnetic resonance spectroscopy (1H-NMR) to confirm their successful synthesis. The 1H-NMR analyses were logged on a Bruker DRX-600 Avance III spectrometer with room temperature deuterated chloroform (CDCl3) as the solvent. DHA, 8arm-PEG-DHA and TF-8arm-PEG-DHA (1 mg of each) were dissolved in 1 mL of CDCl3 and the products were evaluated using an NMR spectrometer”

  • “3.3. Entrapment efficiency and in vitro release study”-this is subheadline

  • “Figure 5. Drug release profile of minocycline loaded Albumin Nanoparticle. Values shown here represent SD (n=3). Two-way Anova, Multiple comparison analysis ***P <0.001.
  • Figure 12. (a)Histopathological evaluation of the major organs of rats treated with physiological saline, minocycline, non-targeted MANP, or targeted tfMANP. Histological analysis of the organs (liver, kidneys, lungs, and spleen) compared to the control group. No abnormal histopathological findings were observed in the liver, kidneys, lungs, or spleen; (b) Body weight changes in blast TBI rats treated with physiological saline, blast, minocycline, non-targeted MANP, or targeted tfMANP”

  • “Because of the formulation of Minocycline as a nanoparticle, we observed decreased accumulation of the drug in the liver, spleen, kidney and heart resulting in the lowering of the toxic effects of minocycline in these tissues [112].”

Reviewer 2 Report

I appreciate the authors presenting this research article. My comments are as follows:

1. Fig 7 In Liver and brain have higher concentrations than others can this be explained

2. It is not clear whether the quantification of particle concentration in Fig 8a. Can it be expressed as a quantitative graph?

3. Fig 12. Is it reasonable to use body weight changes as a proxy for drug toxicity?

4. The limitations of current study?

Author Response

We sincerely thank you for your comments on our manuscript entitled “Enhanced Targeted delivery of minocycline via transferrin conjugated Albumin Nanoparticle and improves neuroprotection in a Blast Traumatic Brain Injury Model”. These comments are all valuable and helpful for us to revise and to improve our manuscript. We have evaluated the comments carefully and have made the necessary corrections. We hope this will meet with your approval. Revised portions are marked in the manuscript as track change and the main corrections and the response to reviewer’s comments are listed below.

We sincerely thank the reviewer for acknowledging our research work. Yes, we have made necessary corrections throughout the manuscript for English language corrections using 3 different experts in English writings.

I appreciate the authors presenting this research article. My comments are as follows:

We sincerely thank the reviewer for acknowledging our research work.

  1. Fig 7 In Liver and brain have higher concentrations than others can this be explained

Thanks for reviewer valuable comments. We have added the following discussion.

The endothelial barrier defines distribution into organs. The nanoparticles extravasate primarily by convection, with its relative influence increasing with transcapillary volume flow. The magnitude of crossing the endothelial barrier is reliant on the capillary pore size, which commonly forms the upper particle size limit. Based on the organ or tissue, the endothelial gap junctions differ from very tight-fisted (blood brain–barrier) to constantly small (standard arteries and capillaries) and fenestrated (digestive mucosa). The liver, as well as lymphoid tissue and hematopoietic organs, comprises sinusoidal (discontinuous) capillaries. Unlike fenestrations, these capillaries lack a diaphragm over the pore and thus show even greater permeability, allowing exchange of nanoparticles. The delivery systems have an intrinsic predilection to the liver. This refers to lipid-based particles, positive and neutral charged nanoparticles, and lipophilic ligands in bioconjugates. Liver accumulation is caused by the sporadic nature of the hepatic vasculature (for nanoparticles and protein bound oligonucleotides).  

  1. It is not clear whether the quantification of particle concentration in Fig 8a. Can it be expressed as a quantitative graph?

Nanoparticle localization is expressed as qualitative image using confocal microscopy. We have also shown biodistribution as quantitative values in brain and other tissues. In future studies, we would be able to calculate the amount using intensity calculation to show the quantitative values.

  1. Fig 12. Is it reasonable to use body weight changes as a proxy for drug toxicity?

Only measuring the body weight would not confirm the toxicity of the substance. It required morphological observation and tissue analysis including the blood. We have also provided histological H&E staining analysis of different organs in addition to body weight analysis. Body weight reduction is generally most pronounced at the end of the studies, the question arises to the concentration and duration determines the strength of effect and how this might be of relevance for the risk assessment. Only those substances were selected which caused a dose-dependent decrease of at least 10–20% in body weight are considered toxic.

  1. The limitations of current study?

Current preliminary study used only IHC for the biochemical analysis. Future studies are aimed to performed detailed molecular and biochemical, behavioral, and chronic toxicity studies which is lacking in the current study.

Furthermore 3 days treatment course was performed and analyzed for 35 days for neuroprotection. But the study needs to evaluate neuroprotection of nanoparticle extended period.

In terms of English correction and plagiarism Ithenticate plagiarism “Some of the technical (material and methods) Even though we have already cited reference articles, plagiarism software is identifying the whole material and methods sections as plagiarism that we are unable to revise or rephrase. Ithenticate plagiarism software also picks up (that amount to total 32%, revised manuscript should be very less) even individual words in a sentence, and subheadings and fig legends that we are unable rephrase or unable to avoid plagiarism. We would like to give the article a brief overview of “materials and methods” and cited appropriate articles for readers understanding. If you are still concerned, we can cut down whole just cite our previously published article with our details. Hope you will be able to understand.

Further, Scientific terminology (for ex., intravenous inj, neurological disease names like as Alzheimer’s disease, Parkinson’s disease, Huntington’s disease, focal cerebral ischemia, multiple sclerosis, traumatic brain injury and spinal cord injury) and the nouns like liver, spleen kidney and body weight were identified by ithenticate plagiarism software.

Following are some of the examples of Plagiarism identified by ithenticate  that we are unable to rephrase or to avoid.

  • “Animals: Ten-week-old male Sprague-Dawley (Charles River Laboratories) rats at 350± 50 g were used in accordance with protocols approved by Rutgers University In-stitutional Animal Care and Use Committee (IACUC approval: PROTO201900142). The animals were housed with free access to food and water in a 12-h dark-light cycle at 22 °C. Rats were divided into five groups (n=6); sham controls, blast and three treatment groups of animals exposed to a moderate blast of 180 kPa)[18,74].”

  • “Rats were perfused with buffered (0.4 M phosphate buffer, pH 7.6) 4% paraformaldehyde. The brain, liver, lungs, kidneys, and spleen, were removed from each animal and immersion-fixed in the same fixative for 24 h at room temperature”.

  • “Statistical analysis

Data are presented as mean ± standard error of the mean (SEM). Between-group comparisons were made by one-way and Two-way analysis of variance (ANOVA) mixed model design with a post hoc test (Tukey) to determine individual group differences. Differences between means were assessed at the probability level of p ≤0.05, 0.01, and 0.001. GraphPad Prism 6.0 software was used in all analyses and preparation of plots”.

  • “Minocycline exhibited characteristic bands at 3487 cm−1. The additional bands at 1597 and 1473 cm−1 are due to the structural vibrations of the benzene rings[97]. The absorption band at 1042 cm−1 was due to C–O stretching vibrations. MANP and tfMANP showed characteristic bands at 3466 and 1653 cm−1 related to the O–H alcohol and C=O groups of acids[98].”

  • “The BSA, Transferrin, minocycline, and tfMANP conjugates were then characterized by proton nuclear magnetic resonance spectroscopy (1H-NMR) to confirm their successful synthesis. The 1H-NMR analyses were logged on a Bruker DRX-600 Avance III spectrometer with room temperature deuterated chloroform (CDCl3) as the solvent. DHA, 8arm-PEG-DHA and TF-8arm-PEG-DHA (1 mg of each) were dissolved in 1 mL of CDCl3 and the products were evaluated using an NMR spectrometer”

  • “3.3. Entrapment efficiency and in vitro release study”-this is subheadline

  • “Figure 5. Drug release profile of minocycline loaded Albumin Nanoparticle. Values shown here represent SD (n=3). Two-way Anova, Multiple comparison analysis ***P <0.001.
  • Figure 12. (a)Histopathological evaluation of the major organs of rats treated with physiological saline, minocycline, non-targeted MANP, or targeted tfMANP. Histological analysis of the organs (liver, kidneys, lungs, and spleen) compared to the control group. No abnormal histopathological findings were observed in the liver, kidneys, lungs, or spleen; (b) Body weight changes in blast TBI rats treated with physiological saline, blast, minocycline, non-targeted MANP, or targeted tfMANP”

  • “Because of the formulation of Minocycline as a nanoparticle, we observed decreased accumulation of the drug in the liver, spleen, kidney and heart resulting in the lowering of the toxic effects of minocycline in these tissues [112].”

Reviewer 3 Report

The paper entitled "Enhanced Targeted delivery of minocycline via transferrin conjugated Albumin Nanoparticle and neuroprotection in Blast Traumatic Brain Injury Model" is very interesting. This paper deals to perform formulation, and characterization of transferrin receptor-targeted conjugated minocycline-loaded albumin nanoparticles in a rat blast TBI model.  The article is well written and only minor spelling English language corrections are required.  I suggest that the authors should improve the discussion about the impact on human uses and the possible collateral effect of the use of transferrin receptor-targeted conjugated minocycline-loaded albumin nanoparticles. Such guidance is believed to increase the quality of the discussion.  

Author Response

We sincerely thank you for your comments on our manuscript entitled “Enhanced Targeted delivery of minocycline via transferrin conjugated Albumin Nanoparticle and improves neuroprotection in a Blast Traumatic Brain Injury Model”. These comments are all valuable and helpful for us to revise and to improve our manuscript. We have evaluated the comments carefully and have made the necessary corrections. We hope this will meet with your approval. Revised portions are marked in the manuscript as track change and the main corrections and the response to reviewer’s comments are listed below.

We sincerely thank the reviewer for acknowledging our research work. Yes, we have made necessary corrections throughout the manuscript for English language corrections using 3 different experts in English writings.

Reviewer 2

The paper entitled "Enhanced Targeted delivery of minocycline via transferrin conjugated Albumin Nanoparticle and neuroprotection in Blast Traumatic Brain Injury Model" is very interesting. This paper deals to perform formulation, and characterization of transferrin receptor-targeted conjugated minocycline-loaded albumin nanoparticles in a rat blast TBI model.  The article is well written and only minor spelling English language corrections are required.  I suggest that the authors should improve the discussion about the impact on human uses and the possible collateral effect of the use of transferrin receptor-targeted conjugated minocycline-loaded albumin nanoparticles. Such guidance is believed to increase the quality of the discussion.  

We sincerely thank the reviewer for acknowledging our research work. Yes, we have made necessary corrections throughout the manuscript using 3 different experts in English writings.

We have also included (Line 143-146) the use of albumin-based nanoparticle and minocycline in human use for treating breast cancer. This will also have potential application in brain injury and other neurological diseases. Nanoparticle formulated using albumin was approved by FDA for pancreatic cancer, breast cancer, and non-small cell lung cancer i.e., Abraxane (protein-bound paclitaxel) which showed improved overall survival in 861 patients with metastatic pancreatic cancer. 

In terms of English correction and plagiarism Ithenticate plagiarism “Some of the technical (material and methods) Even though we have already cited reference articles, plagiarism software is identifying the whole material and methods sections as plagiarism that we are unable to revise or rephrase. Ithenticate plagiarism software also picks up (that amount to total 32%, revised manuscript should be very less) even individual words in a sentence, and subheadings and fig legends that we are unable rephrase or unable to avoid plagiarism. We would like to give the article a brief overview of “materials and methods” and cited appropriate articles for readers understanding. If you are still concerned, we can cut down whole just cite our previously published article with our details. Hope you will be able to understand.

Further, Scientific terminology (for ex., intravenous inj, neurological disease names like as Alzheimer’s disease, Parkinson’s disease, Huntington’s disease, focal cerebral ischemia, multiple sclerosis, traumatic brain injury and spinal cord injury) and the nouns like liver, spleen kidney and body weight were identified by ithenticate plagiarism software.

Following are some of the examples of Plagiarism identified by ithenticate  that we are unable to rephrase or to avoid.

  • “Animals: Ten-week-old male Sprague-Dawley (Charles River Laboratories) rats at 350± 50 g were used in accordance with protocols approved by Rutgers University In-stitutional Animal Care and Use Committee (IACUC approval: PROTO201900142). The animals were housed with free access to food and water in a 12-h dark-light cycle at 22 °C. Rats were divided into five groups (n=6); sham controls, blast and three treatment groups of animals exposed to a moderate blast of 180 kPa)[18,74].”

  • “Rats were perfused with buffered (0.4 M phosphate buffer, pH 7.6) 4% paraformaldehyde. The brain, liver, lungs, kidneys, and spleen, were removed from each animal and immersion-fixed in the same fixative for 24 h at room temperature”.

  • “Statistical analysis

Data are presented as mean ± standard error of the mean (SEM). Between-group comparisons were made by one-way and Two-way analysis of variance (ANOVA) mixed model design with a post hoc test (Tukey) to determine individual group differences. Differences between means were assessed at the probability level of p ≤0.05, 0.01, and 0.001. GraphPad Prism 6.0 software was used in all analyses and preparation of plots”.

  • “Minocycline exhibited characteristic bands at 3487 cm−1. The additional bands at 1597 and 1473 cm−1 are due to the structural vibrations of the benzene rings[97]. The absorption band at 1042 cm−1 was due to C–O stretching vibrations. MANP and tfMANP showed characteristic bands at 3466 and 1653 cm−1 related to the O–H alcohol and C=O groups of acids[98].”

  • “The BSA, Transferrin, minocycline, and tfMANP conjugates were then characterized by proton nuclear magnetic resonance spectroscopy (1H-NMR) to confirm their successful synthesis. The 1H-NMR analyses were logged on a Bruker DRX-600 Avance III spectrometer with room temperature deuterated chloroform (CDCl3) as the solvent. DHA, 8arm-PEG-DHA and TF-8arm-PEG-DHA (1 mg of each) were dissolved in 1 mL of CDCl3 and the products were evaluated using an NMR spectrometer”

  • “3.3. Entrapment efficiency and in vitro release study”-this is subheadline

  • “Figure 5. Drug release profile of minocycline loaded Albumin Nanoparticle. Values shown here represent SD (n=3). Two-way Anova, Multiple comparison analysis ***P <0.001.
  • Figure 12. (a)Histopathological evaluation of the major organs of rats treated with physiological saline, minocycline, non-targeted MANP, or targeted tfMANP. Histological analysis of the organs (liver, kidneys, lungs, and spleen) compared to the control group. No abnormal histopathological findings were observed in the liver, kidneys, lungs, or spleen; (b) Body weight changes in blast TBI rats treated with physiological saline, blast, minocycline, non-targeted MANP, or targeted tfMANP”

  • “Because of the formulation of Minocycline as a nanoparticle, we observed decreased accumulation of the drug in the liver, spleen, kidney and heart resulting in the lowering of the toxic effects of minocycline in these tissues [112].”

Reviewer 4 Report

-Abbreviations should be uniform and provided with the first mention in the text (for example, Tf is also tf )

-English language must be improved

- There are some unnecessary pieces of information which made the manuscript harder to read

-Problematic spacing between words

Line 19,20, 24,25: The sentences are not well written and not understandable

Line 45: The authors are from different countries, "in acts of terrorism and warfare domestically and abroad" should be "in acts of terrorism and warfare"

Line 51: The sentence is interrupted, plus problematic spacing

Figure 1 is narrow, and not easy to read, some abbreviations are used without introducing them so it is unclear what is all about (mANP)

-600 sec is 10 min, if everything is written in min, this should not be left in sec

-What were the objects in NOR?

-First part of the discussion should be moved to Material and Methods

-Inconsistency with abbreviations, it seems that several people were writing it with everybody introducing abbreviations and there was not final reading of the manuscript

-Figure or Fig?

-Line 454: (M-ANP) or MANP?

Author Response

We sincerely thank you for your comments on our manuscript entitled “Enhanced Targeted delivery of minocycline via transferrin conjugated Albumin Nanoparticle and improves neuroprotection in a Blast Traumatic Brain Injury Model”. These comments are all valuable and helpful for us to revise and to improve our manuscript. We have evaluated the comments carefully and have made the necessary corrections. We hope this will meet with your approval. Revised portions are marked in the manuscript as track change and the main corrections and the response to reviewer’s comments are listed below.

We sincerely thank the reviewer for acknowledging our research work. Yes, we have made necessary corrections throughout the manuscript for English language corrections using 3 different experts in English writings.

  1. -Abbreviations should be uniform and provided with the first mention in the text (for example, Tf is also tf )

We have made corrections in the abbreviation tf looks uniform throughout the manuscript.

  1. -English language must be improved
    • There are some unnecessary pieces of information which made the manuscript harder to read. Problematic spacing between words.

Yes, we have made necessary corrections throughout the manuscript for English language corrections using 3 different experts in English writings. Removed unnecessary contents as well as shown in track change in the manuscript. Corrections made as suggested by the reviewer.

  1. Line 19,20, 24,25: The sentences are not well written and not understandable

These sentences are revised and the spacing adjusted as suggested by the reviewer.

  1. Line 45: The authors are from different countries, "in acts of terrorism and warfare domestically and abroad" should be "in acts of terrorism and warfare".

The sentence is revised and the spacing adjusted as suggested by the reviewer.

  1. Line 51: The sentence is interrupted, plus problematic spacing

The sentence is revised and the spacing adjusted as suggested by the reviewer.

  1. Figure 1 is narrow, and not easy to read, some abbreviations are used without introducing them so it is unclear what is all about (mANP).

Better resolution images are added, and the abbreviation details included in the legend as minocycline loaded transferrin conjugated albumin nanoparticle (tf-MANP).

  1. -600 sec is 10 min, if everything is written in min, this should not be left in sec

Corrections made as suggested by the reviewer.

  1. -What were the objects in NOR?

In this study we used objects like lego bricks, transparent plastic bottle with sand, and glass bottles with water.

  1. -First part of the discussion should be moved to Material and Methods

Corrections made as suggested by the reviewer.

  1. -Inconsistency with abbreviations, it seems that several people were writing it with everybody introducing abbreviations and there was not final reading of the manuscript. -Figure or Fig? -Line 454: (M-ANP) or MANP?

Corrections made as suggested by the reviewer. It looks uniform as MANP throughout manuscript.

In terms of English correction and plagiarism Ithenticate plagiarism “Some of the technical (material and methods) Even though we have already cited reference articles, plagiarism software is identifying the whole material and methods sections as plagiarism that we are unable to revise or rephrase. Ithenticate plagiarism software also picks up (that amount to total 32%, revised manuscript should be very less) even individual words in a sentence, and subheadings and fig legends that we are unable rephrase or unable to avoid plagiarism. We would like to give the article a brief overview of “materials and methods” and cited appropriate articles for readers understanding. If you are still concerned, we can cut down whole just cite our previously published article with our details. Hope you will be able to understand.

Further, Scientific terminology (for ex., intravenous inj, neurological disease names like as Alzheimer’s disease, Parkinson’s disease, Huntington’s disease, focal cerebral ischemia, multiple sclerosis, traumatic brain injury and spinal cord injury) and the nouns like liver, spleen kidney and body weight were identified by ithenticate plagiarism software.

Following are some of the examples of Plagiarism identified by ithenticate  that we are unable to rephrase or to avoid.

  • “Animals: Ten-week-old male Sprague-Dawley (Charles River Laboratories) rats at 350± 50 g were used in accordance with protocols approved by Rutgers University In-stitutional Animal Care and Use Committee (IACUC approval: PROTO201900142). The animals were housed with free access to food and water in a 12-h dark-light cycle at 22 °C. Rats were divided into five groups (n=6); sham controls, blast and three treatment groups of animals exposed to a moderate blast of 180 kPa)[18,74].”

  • “Rats were perfused with buffered (0.4 M phosphate buffer, pH 7.6) 4% paraformaldehyde. The brain, liver, lungs, kidneys, and spleen, were removed from each animal and immersion-fixed in the same fixative for 24 h at room temperature”.

  • “Statistical analysis

Data are presented as mean ± standard error of the mean (SEM). Between-group comparisons were made by one-way and Two-way analysis of variance (ANOVA) mixed model design with a post hoc test (Tukey) to determine individual group differences. Differences between means were assessed at the probability level of p ≤0.05, 0.01, and 0.001. GraphPad Prism 6.0 software was used in all analyses and preparation of plots”.

  • “Minocycline exhibited characteristic bands at 3487 cm−1. The additional bands at 1597 and 1473 cm−1 are due to the structural vibrations of the benzene rings[97]. The absorption band at 1042 cm−1 was due to C–O stretching vibrations. MANP and tfMANP showed characteristic bands at 3466 and 1653 cm−1 related to the O–H alcohol and C=O groups of acids[98].”

  • “The BSA, Transferrin, minocycline, and tfMANP conjugates were then characterized by proton nuclear magnetic resonance spectroscopy (1H-NMR) to confirm their successful synthesis. The 1H-NMR analyses were logged on a Bruker DRX-600 Avance III spectrometer with room temperature deuterated chloroform (CDCl3) as the solvent. DHA, 8arm-PEG-DHA and TF-8arm-PEG-DHA (1 mg of each) were dissolved in 1 mL of CDCl3 and the products were evaluated using an NMR spectrometer”

  • “3.3. Entrapment efficiency and in vitro release study”-this is subheadline

  • “Figure 5. Drug release profile of minocycline loaded Albumin Nanoparticle. Values shown here represent SD (n=3). Two-way Anova, Multiple comparison analysis ***P <0.001.
  • Figure 12. (a)Histopathological evaluation of the major organs of rats treated with physiological saline, minocycline, non-targeted MANP, or targeted tfMANP. Histological analysis of the organs (liver, kidneys, lungs, and spleen) compared to the control group. No abnormal histopathological findings were observed in the liver, kidneys, lungs, or spleen; (b) Body weight changes in blast TBI rats treated with physiological saline, blast, minocycline, non-targeted MANP, or targeted tfMANP”

  • “Because of the formulation of Minocycline as a nanoparticle, we observed decreased accumulation of the drug in the liver, spleen, kidney and heart resulting in the lowering of the toxic effects of minocycline in these tissues [112].”

Round 2

Reviewer 1 Report

manuscript can be accepted in current revised format